# Dealing with Uncertainty in Contextual Anomaly Detection

**Luca Bindini**                                                    *luca.bindini@unifi.it*
*AI Lab, Department of Information Engineering*
*University of Florence, Italy*

**Lorenzo Perini**                                           *lorenzo.perini@kuleuven.be*
*DTAI Research Unit, Department of Computer Science & Leuven.AI*
*KU Leuven, Belgium*

**Stefano Nistri**                                           *stefanonistri41@gmail.com*
*Cardiology Service*
*CMSR Veneto Medica, Italy*

**Jesse Davis**                                                *jesse.davis@kuleuven.be*
*DTAI Research Unit, Department of Computer Science & Leuven.AI*
*KU Leuven, Belgium*

**Paolo Frasconi**                                             *paolo.frasconi@unifi.it*
*AI Lab, Department of Information Engineering*
*University of Florence, Italy*

**Reviewed on OpenReview:** *https://openreview.net/forum?id=yLoXQDNwwa*

## Abstract

Contextual anomaly detection (CAD) aims to identify anomalies in a target (behavioral) variable *conditioned* on a set of contextual variables that influence the normalcy of the target variable but are not themselves indicators of an anomaly. In this work, we propose a novel framework for CAD, *normalcy score* (NS), that explicitly models both the aleatoric and epistemic uncertainties. Built on heteroscedastic Gaussian process regression, our method regards the Z-score as a random variable, providing a contextual anomaly score but also a high-density interval that reflect the reliability of the contextual anomaly assessment. Through experiments on benchmark datasets and a real-world application in cardiology, we demonstrate that NS outperforms state-of-the-art CAD methods in both detection accuracy and interpretability. Moreover, high-density intervals enable an adaptive, uncertainty-driven decision-making process, which may be very important in domains such as healthcare.

## 1 Introduction

In some applications of anomaly detection, there are variables that only play a contextual role and are not indicators of an object being anomalous. As a running example for this paper, suppose that we are interested in assessing the normalcy of aorta diameters in patients monitored for potential cardiac risk. In this application, we need to collect contextual variables such as weight, height, sex, and age, since the size of aorta (the variable for which we are genuinely trying to assess normalcy) depends on them. However, in this specific application, we do not want to flag patients as anomalous if they are overweight or if they are too short for their age.

This particular problem setting is known as conditional (or contextual) anomaly detection (CAD) (Song et al., 2007; Valko et al., 2011; Tang et al., 2013; Liang & Parthasarathy, 2016; Li & Van Leeuwen, 2023). The name originates from the simple observation that rather than looking at the joint distribution of all

variables, we split variables in two groups: contextual variables, $x$, and behavioral variables[1], $y$. We then fit a conditional model and say that a given $x$ is anomalous if $p(y|x)$ is small according to our model. Note that modeling the conditional rather than the joint distribution is statistically more efficient, allowing us to work in application domains where data is not abundant.

Related techniques started to be developed much earlier in medical statistics for constructing reference intervals (RIs) and centile curves (Goldstein, 1972; Healy, 1978; Royston, 1991; Altman, 1993; Zierk et al., 2021; Ammer et al., 2023). One example application is the construction of child grow standards that provide percentiles of height and weight stratified by age and sex (WHO Multicentre Growth Reference Study Group, 2006). In this body of literature (which, interestingly, does not intersect with the more recent machine learning CAD literature), contextual variables $x$ (such as age and sex) play the role of covariates while $y$ (e.g., height) plays the role of a response variable and is typically a continuous scalar. A classic approach is to assume that $y|x$ (perhaps after performing some variable transformations (Box & Cox, 1964)) is normally distributed, fit a regression model to predict $y$ from $x$, and finally compute a Z-score (Roman et al., 1989; Lee et al., 2010; Colan, 2013) as the difference between predictions and measurement, scaled by error standard deviation (see also Section 2). A Z-score above 2 is a typical threshold to flag an anomaly.

A common trait of both CAD and methods for constructing RIs is that anomalous (out-of-reference) data points are identified by analyzing the conditional distribution $p(y|x)$. Effectively, these approaches are trying to model the aleatoric uncertainty (AU), which originates from the intrinsic variability of the behavior within the subpopulation that share the same context. The (implicit) underlying assumption is that anomalous data points are exactly those with large AU. From the point of view of the model, there is no way of reducing AU, except maybe by including more contextual variables (which may not even be possible). However, there is another source of uncertainty: we may lack samples in certain regions of the data space. This is known as *epistemic* uncertainty (EU) (Senge et al., 2014; Depeweg et al., 2018; Hüllermeier & Waegeman, 2021), and in the present setting it occurs when a certain context is underrepresented in the data. On the one hand EU may arise because certain subpopulations are inherently (very) small and not included in the data. On the other hand, EU may arise due to various ethical, practical, and logistic challenges, which is the case in medical domains such as pediatrics (Ammer et al., 2023; Zierk et al., 2021; Ceriotti, 2012), or due to various exclusion criteria, which is the case for imaging studies in cardiology (Ricci et al., 2021; Asch et al., 2019; Lancellotti et al., 2013). In our running example, overweight children may not appear in the data for these reasons. Accounting for the EU is crucial because normalcy assessment may be *unreliable* if the model's EU is large.

Despite the importance of modeling both AU and EU, to the best of our knowledge neither existing CAD nor normalcy approaches in medicine distinguish between them. This paper fills this gap. Specifically, we instantiate this idea as heteroscedastic Gaussian process regression (HGPR) (Goldberg et al., 1997) that uses separate Gaussian processes to model the mean and standard deviation. This approach confers several advantages and overcomes some well-known limitations of classic Z-scores (see, e.g., (Mawad et al., 2013; Colan, 2013; Curtis et al., 2016)). First, it models the fact that the variability of behavior depends on the context. Consequently, what is considered anomalous behavior is coupled to the specific context. Second, using two Gaussian processes enables disentangling the AU in the data and EU in the model. This idea has been previously explored for active learning (Patel et al., 2022) but not for anomaly detection. Third, being a Bayesian approach, it regards $Z$ as a random variable and thus naturally provides a method for computing high-density intervals on the anomaly score. This allows conveying a sense of the uncertainty of the model's assigned anomaly score.

Our key contributions are:

- We explicitly disentangle AU and EU with two independent Gaussian processes.

- We enrich the anomaly score with a 95 % Highest-Density Interval (HDI), providing a calibrated and interpretable measure.

---

[1]In Song et al. (2007), contextual variables are called environmental attributes and behavioral variables are called indicator attributes.

- We validate our method on standard publicly available benchmarks, consistently outperforming established CAD and AD methods in terms of ROC AUC and PR AUC.

- We demonstrate the effectiveness of our approach on a real-world clinical cohort involving aortic measurements, showcasing its potential impact in cardiology.

## 2 Related Work

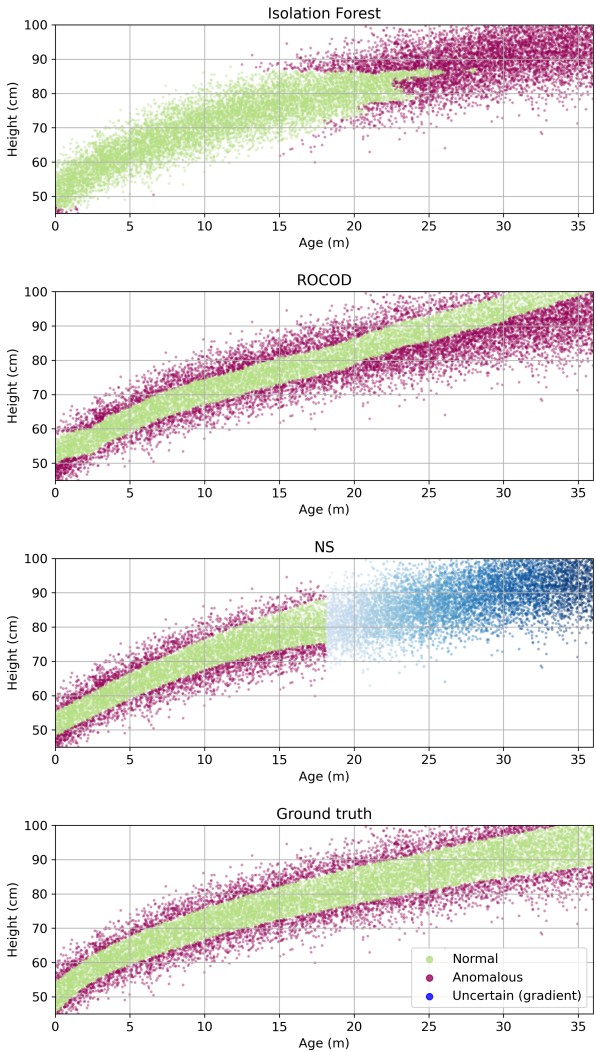

Figure 1: Simulated data from WHO growth curves for girls. Here $H \sim \mathcal{N}(\overline{h}(A), \overline{\sigma}(A))$, where the age-indexed mean function $\overline{h}(A)$ and standard deviation $\overline{\sigma}(A)$ are derived from published data (WHO Multicentre Growth Reference Study Group, 2006). In the training data, $A \sim \mathrm{Exp}(0.4)$, so that young girls have a much higher probability of appearing in the training data. Non-contextual anomaly detection algorithms (such as isolation forests) trained to find the support of the joint $p(H, A)$ fail to spot children that are too tall or too short for their age, and additionally flag as outliers data points where the marginal $p(A)$ is small. ROCOD, a state-of-the-art algorithm for CAD, does not suffer from this problem but fails to correctly estimate conditional outliers in the presence of context-specific AU (heteroscedasticity). Our approach (NS) correctly identifies conditional outliers and, in addition, can abstain when EU (estimated as the width of the 95% HDI, $i(H, A)$, see Section 3.2) is high: In the plot, points with $i(H, A) > 2$ are colored in blue with an intensity proportional to $i(H, A)$).

The task of contextual anomaly detection (CAD) can be formalized as follows. Let $\mathcal{D} = (x_i, y_i)_{i=1}^N$ be a dataset, where $x_i \in \mathbb{R}^P$ are contextual variables and $y_i \in \mathbb{R}$ are behavioral variables. The goal of CAD is to identify data points that are anomalous with respect to the conditional distribution $p(y|x)$. Formally, CAD involves learning a probabilistic model for the conditional distribution $p(y|x)$ that assigns an anomaly score $s(x, y)$ to each observation $(x, y)$, where lower probabilities correspond to higher anomaly scores. This approach ensures that points in high-probability regions of $p(y|x)$ are assigned low anomaly scores, aligning with the intuition that less probable points should be considered more anomalous.

The foundational work on CAD was introduced in Song et al. (2007). In that paper, contextual and behavioral distributions were modeled independently using Gaussian mixture models and their dependencies were learned through mapping functions. Anomalies can be then identified as deviations from the learned dependencies. However, the approach may overstimate outlierness of objects with unusual contexts, as it would happen by modeling the joint $p(x, y)$ (see also Figure 1).

ROCOD (Liang & Parthasarathy, 2016) aims to mitigate the context sparsity problem by introducing a dual modeling approach that leverages both local and global perspectives to detect anomalies. The local model in ROCOD estimates the expected behavior of an object based on its contextual neighbors, which are identified as the objects most similar in terms of contextual attributes. In contrast, the global model adopts a holistic view of the dataset, learning the overall dependencies between contextual and behavioral variables through regression models. These two models are integrated via an adaptive weighting mechanism. ROCOD dynamically adjusts the relative importance of the local and global models based on the density of contextual neighbors for each object. When an object has a sufficient number of contextual neighbors, the local model is prioritized. Conversely, in sparse contexts, the global model dominates, ensuring robust anomaly detection across diverse scenarios.

QCAD (Li & Van Leeuwen, 2023) builds upon CAD and ROCOD by using quantile regression forests (Meinshausen & Ridgeway, 2006) to model $p(y|x)$. Unlike ROCOD, which relies on the conditional mean to infer expected behavior, QCAD estimates conditional quantiles, providing a more detailed representation of the underlying data distribution. The anomaly score is computed by estimating the deviation of the observed $y$ from the predicted $p(y|x)$, looking at the width of the conditional percentile interval in which $y$ falls. Intuitively, if $y$ lies outside the estimated quantile range, the score is adjusted proportionally.

Due to its simplicity and interpretability, the Z-score is widely used in normalcy studies (Roman et al., 1989; Colan, 2013; Campens et al., 2014) and in the development of reference intervals (Lee et al., 2010; Zierk et al., 2021). In our context, the classic Z-score is defined as follows:

$$Z(x, y) = \frac{y - f(x)}{S} \tag{1}$$

where $y$ and $x$ are the measured values of the behavioral variable and the contextual variables, respectively, $f$ is a model trained to predict behavior from context, and $S$ an estimation of the standard deviation (such as the square root of the square loss on the training set). Often, $f$ is a linear model, perhaps after the application of some ad-hoc Box-Cox transformations (Box & Cox, 1964) (for example, one might regress the log of height on age in Figure 1 in order to model the saturation of growth).

The above Z-score definition does not model heteroscedasticity in the data, i.e., situations when variability of the behavioral variable in the populations depends on the given context. Since the denominator averages out this variability, $Z(x, y)$ is overestimated in low-variance contexts and underestimated in high-variance contexts (Chubb & Simpson, 2012; Mawad et al., 2013). Early approaches attempted to reduce heteroscedasticity by splitting data into separate groups (Goldstein, 1972). Later, Altman (1993) proposed a way to estimate variance by training a second linear regression model on the absolute residuals of the first model, a technique that may be seen as a simplified, linear, and non-Bayesian version of the technique proposed in this paper.

## 3   Methodology

This section describes our proposed *normalcy score* (NS), which tailors the classic Z-score to the contextual anomaly detection setting. Unlike the Z-score, NS is a random variable, even when the model and the test point $(x, y)$ are fixed. As in Le et al. (2005), we *jointly* train two Gaussian process regression (GPR) models: one to model the mean and one to model the variance of the target variable as functions of the contextual features. We use GPR for three reasons. First, it provides a flexible non-parametric framework for modeling $p(y|x)$ as a Gaussian distribution. Intuitively, GPR defines a prior over functions $f(x)$ with variance $\sigma^2(x)$ using a mean function $m(x)$ and a covariance (kernel) function $k(x, x')$, which encodes assumptions about smoothness and relationships between data points. Second, it allows disentangling the aleatoric and epistemic uncertainty. Finally, it permits providing both a point estimate and a high-density interval for an object's anomaly score.

We begin by formalizing the definition of NS and its underlying Bayesian framework; then we describe how high-density intervals are derived for anomaly scores.

### 3.1   Normalcy Score

NS quantifies the deviation of a target variable $y$ from its expected behavior relative to the contextual variance. To define NS, we first introduce two key components modeled using Gaussian process regression (GPR): $f_1$ and $f_2$.

The first GP models the mean of the target variable $y$ given the contextual features $x$:

$$f_1(x) \sim \mathcal{GP}(m_1(x), k_1(x, x')),$$

where $m_1(x)$ is the mean function, and $k_1(x, x')$ is the covariance function of the GP.

The second GP models the log standard deviation:

$$f_2(x) \sim \mathcal{GP}(m_2(x), k_2(x, x')),$$

where $m_2(x)$ is the mean function, and $k_2(x, x')$ is the covariance function.

We model the log-standard deviation to ensure that the predicted standard deviation remains strictly positive following Goldberg et al. (1997).

With these definitions, NS is given by:

$$\text{NS}(x, y) = \frac{y - f_1(x)}{e^{f_2(x)}},$$

where $y$ is the observed value of the target variable. Note that $\text{NS}(x, y)$ is defined similarly to $Z(x, y)$ but is a random variable.

A key challenge with standard GPs is their running time, which scales as $O(N^3)$, where $N$ is the number of points in the dataset. To address this, we use a sparse approximation based on inducing variables (Snelson & Ghahramani, 2005; Titsias, 2009). Using a set of $M$ inducing points to summarize the GP posterior and approximate the covariance matrix, the running time is reduced to $O(M^2 N)$, where $M \ll N$.

We use NS to disentangle aleatoric uncertainty (AU) and epistemic uncertainty (EU) as follows:

- AU: Captured directly by the second GP $f_2$, which models the log standard deviation. This uncertainty reflects the intrinsic variability of $y$ given $x$, which cannot be reduced even with more data.

- EU: Captured through the posterior of both $f_1$ and $f_2$. Sparse regions in the contextual feature space lead to higher posterior variances, as the GPs are less certain about the function values in those areas. By sampling from the posterior distributions of $f_1$ and $f_2$, we can quantify this uncertainty.

To summarize:

- We place independent GP priors on $f_1$ and $f_2$, modeling respectively the mean and the log standard deviation of $y \mid x$.

- To make inference scalable, we adopt a sparse variational approximation with inducing points.

- Training is performed using natural gradient descent for the variational parameters and the Adam optimizer for the remaining parameters.

### 3.2 Characterizing the NS distribution

Our model assumes that both $f_1$ and $f_2$ are drawn from Gaussian processes. Specifically, the residual $y - f_1(x)$ follows a Gaussian distribution. Separately, the reciprocal of the standard deviation term $e^{-f_2(x)}$, where $f_2(x)$ models the log standard deviation, follows a log-normal distribution:

$$e^{-f_2(x)} \sim \mathrm{LogNormal}(-m_2(x), \sigma_2^2(x)).$$

As a result, $p(\mathrm{NS}(x, y))$ is not normal. Still, we wish to summarize it in terms of position and dispersion, that quantify AU and EU, respectively.

**Computing a point estimate.** The score $s(x, y)$ for a given object is simply defined as the expected value of NS. The independence of the two processes allows us to compute it as follows:

$$s(x, y) \doteq \mathbb{E}[\mathrm{NS}(x, y)] = \mathbb{E}[y - f_1(x)] \cdot \mathbb{E}[e^{-f_2(x)}].$$

Using properties of the log-normal distribution, this simplifies to:

$$s(x, y) = (y - m_1(x))e^{-m_2(x) + \sigma_2^2(x)/2}.$$

**Computing the high-density interval.** One key aspect of the Bayesian approach is that it offers a principled method for quantifying the EU associated with the predicted mean and scale by looking at the concentration of the density $p(\mathrm{NS}(x, y))$. A simple way of expressing the concentration of $p(\mathrm{NS}(x, y))$ is by computing its *highest density interval* (HDI) (Kruschke, 2015), defined as the smallest interval of values of NS such that $p(\mathrm{NS}(x, y))$ integrates to a given threshold (usually 95%). There is no general analytic form for the HDI. Hence, we first obtain samples of NS by sampling from $f_1$ and $f_2$ and then estimate $p$ by kernel density estimation. We finally perform numerical integration. We denote the length of the 95% interval as $i(x, y)$. As the dataset size increases, EU decreases and $p(\mathrm{NS}(x, y))$ becomes more concentrated, resulting in a smaller HDI.

**Aleatoric and epistemic uncertainty.** As more data accumulate in the neighbourhood of $x$, the posterior means $m_1(x)$ and $m_2(x)$ converge to the underlying regression functions, while the posterior variances $\sigma_1^2(x)$ and $\sigma_2^2(x)$ shrink. This captures the reduction of epistemic uncertainty with increasing sample size. In contrast, the aleatoric component $e^{m_2(x)}$, representing the intrinsic variability of $y \mid x$, does not vanish even in the limit of infinite data. The expectation of $\mathrm{NS}(x, y)$ therefore contains a multiplicative correction factor $e^{\sigma_2^2(x)/2}$ arising from the log-normal distribution of $e^{-f_2(x)}$; this term converges to 1 as $\sigma_2^2(x)$ decreases and ensures that the point estimate $s(x, y)$ is consistent with the exact posterior expectation. The length of the high-density interval $i(x, y)$, on the other hand, depends on both $\sigma_1^2(x)$ and $\sigma_2^2(x)$ and shrinks as these posterior variances decrease, making it a faithful indicator of epistemic uncertainty.

## 4 Experiments

The goal of the empirical evaluation is to address the following four questions:

- **RQ1:** How does NS perform in detecting contextual anomalies compared to other methods?

Table 1: Summary of UCI benchmark datasets with injected anomalies.

| Dataset | #Samples | Anomaly Ratio | Behavioral Variable |
|---------|----------|---------------|---------------------|
| Abalone | 4177 | 2.4% | Rings |
| Concrete | 1030 | 4.8% | Strength |
| SynMachine | 557 | 8.9% | IF |
| Toxicity | 908 | 5.5% | Toxicity |
| Yacht | 308 | 9.7% | Resistance |

- **RQ2:** How sensitive is the framework to the choice of kernel and number of inducing points?

- **RQ3:** Does $i(x, y)$ correlate with sparsity in contextual feature space?

- **RQ4:** Can NS aid in clinical practice for assessing the normalcy of the aortic diameter?

## 4.1 Data

**Benchmark datasets.** Table 1 summarizes the five UCI benchmark datasets, which include *Abalone* (Nash et al., 1994), containing physical measurements of abalones used to predict their age; *Concrete Compressive Strength* (Yeh, 1998), describing the properties of concrete mixtures with the target being the compressive strength; *Synchronous Machine* (UCI, 2021), representing the behavior of synchronous motors in electrical systems; *QSAR Fish Toxicity* (Ballabio et al., 2015), which assesses the acute toxicity of chemicals on fish; and *Yacht Hydrodynamics* (Gerritsma et al., 1981), which involves predicting the hydrodynamic resistance of sailing yachts based on hull geometry and operating conditions.

Because these datasets do not contain contextual anomalies, we follow the experimental protocol introduced in Li & Van Leeuwen (2023) which injects anomalies into the entire dataset before performing any train/test split. Given a dataset $\{(x_i, y_i), i = 1, \ldots, N\}$, where each context consists of $P$ variables, $x_i = \{x_{i1}, \ldots, x_{iP}\}$, the anomaly injection process modifies the dataset as follows:

1. Normalize the behavioral feature $y$ in the dataset to the range $[0, 1]$ using MinMax scaling, while keeping the contextual features $x = \{x_1, \ldots, x_P\}$ unchanged.

2. Select $n$ instances $(0 < n \ll N)$ randomly from the dataset.

3. For each selected instance $(x_i, y_i)$, perturb the behavioral feature $y_i$ to be $\hat{y}_i = y_i + \epsilon_i$, where $\epsilon_i$ is sampled uniformly from the range $[-0.5, -0.1] \cup [0.1, 0.5]$. The contextual features $x_i$ remain unchanged which ensures that the anomaly arises purely due to the behavioral feature $\hat{y}_i$ deviating from its expected value given its context $x_i$.

4. Replace the original instance $(x_i, y_i)$ with the perturbed instance $(x_i, \hat{y}_i)$.

This scheme integrates the injected anomalies into the dataset while preserving the structure of the contextual features. Moreover, it addresses limitation of Song et al. (2007)'s approach where some injected anomalies remained contextually normal. Table 1 shows the percent of perturbed examples in each dataset.

**Real-world dataset.** We use a medical dataset on aorta normalcy (Frasconi et al., 2021). Aortic dilation is a risky condition with potentially deadly outcomes such as regurgitation or dissection. The dataset consists of 1110 normal subjects, and 351 patients with bicuspid aortic valve. The contextual variables are age, gender, weight, and height. We consider, separately, two behavioral variables, i.e., aorta diameters measured at two different locations: sinus of valsalva (SoV) and ascending aorta (AA).

## 4.2 Experimental Setup

We compare our approach to three types of anomaly detection methods. First, we consider the contextual approaches QCAD and ROCOD, using the implementation from the QCAD repository[2] with the hyperparameter settings as in the original paper (number of trees in the quantile regression forest, maximal features, minimum node size, and number of conditional quantiles). Second, we include regression-based contextual baselines. For the Z-score in Equation 1, we fit a linear regression model on the training set and use a single global residual scale $S$ estimated from the training residuals. We further include the linear variance-estimation approach of Altman (1993) and, to isolate the contribution of heteroscedasticity, a homoscedastic variant of our method, denoted $NS_{hom}$, which uses a single GP. Finally, we use the standard non-contextual anomaly detection algorithms Isolation Forest (IForest) (Liu et al., 2008), Local Outlier Factor (LOF) (Breunig et al., 2000), and Histogram-based Outlier Score (HBOS) (Goldstein & Dengel, 2012) from the PyOD library (Zhao et al., 2019) with their default configurations.

The proposed NS relies on the heteroscedastic Gaussian Process Regressor (HGPR) model, implemented using the GPFlow library (Matthews et al., 2017). Key aspects of the HGPR model configuration include the use of a Rational Quadratic kernel (Rasmussen & Williams, 2006), which is well-suited for capturing both smooth and non-smooth variations in the data. The heteroscedastic prior is modeled with a constant mean 0 and very low constant variance of $10^{-5}$. Inducing variables are initialized to the first 5% of the training data and updated during optimization, with the number of inducing points set proportionally to the dataset size to ensure scalability. The model is trained using a hybrid optimization approach that combines Natural Gradient Descent with $\gamma = 0.02$ and the Adam optimizer with a learning rate of 0.01. We trained models for a maximum of 40,000 epochs.

To evaluate anomaly detection performance, we use a 5-fold cross-validation strategy. This ensures that every data point is evaluated as part of the test set once. We adopt widely accepted metrics in the anomaly detection literature (Liang & Parthasarathy, 2016; Aggarwal, 2017; Kuo et al., 2018; Li & Van Leeuwen, 2023) like area under the ROC curve (ROC AUC) and area under the precision recall curve (PR AUC).

## 4.3 Results

**RQ1: How does NS perform in detecting contextual anomalies compared to other methods?**
Table 2 reports the average over five independent runs (each repeating the injection procedure with a different seed) together with the standard deviation for both metrics across the benchmark datasets. The NS anomaly score is computed as the absolute value of the expected NS, as detailed in Section 3.2, so that deviations in either direction are treated symmetrically.

NS secures the top ROC AUC and PR AUC on *Abalone*, *SynMachine*, *Toxicity*, and *Yacht*, with *Concrete* remaining the lone dataset where QCAD holds a narrow edge. NS attains perfect scores on *SynMachine*; its strong 0.97/0.88 ROC AUC/PR AUC on *Yacht*, together with competitive PR AUC values of 0.65 on *Abalone* and 0.67 on *Toxicity*, underscores the method's robustness across heterogeneous datasets.

Both $NS_{hom}$ and the Altman method deliver strong results on several datasets, while NS remains the most consistently strong approach overall. Nonetheless, it is striking that an off-the-shelf Z-score baseline, overlooked in prior work, already proves remarkably competitive on this benchmark. To the best of our knowledge, this is the first work to include such a Z-score baseline in a contextual anomaly detection benchmark, and its strong results show it should be treated as a meaningful reference point.

Non-contextual methods, including Isolation Forest, LOF, and HBOS, consistently underperform across all datasets and metrics. This result underscores the critical role of explicitly incorporating contextual information into anomaly detection models. Importantly, the gap is driven by the problem formulation. These methods model $p(x, y)$ rather than the conditional $p(y \mid x)$, and the effect is not due to the particular choice of detector. We therefore use standard non-contextual baselines commonly reported in prior CAD studies.

---

[2]https://github.com/ZhongLIFR/QCAD

Overall, the results highlight the robustness and versatility of NS, which consistently performs well across diverse domains and balances performance across diverse evaluation metrics.

Table 2: Average ($\pm$ std) ROC AUC and PR AUC computed across five independent anomaly injections (each evaluated with 5-fold cross-validation).

| Method | Abalone ROC AUC | Abalone PR AUC | Concrete ROC AUC | Concrete PR AUC | SynMachine ROC AUC | SynMachine PR AUC | Toxicity ROC AUC | Toxicity PR AUC | Yacht ROC AUC | Yacht PR AUC |
|---|---|---|---|---|---|---|---|---|---|---|
| NS | **0.96 $\pm$ 0.01** | **0.65 $\pm$ 0.04** | 0.89 $\pm$ 0.02 | 0.60 $\pm$ 0.01 | **1.00 $\pm$ 0.00** | **1.00 $\pm$ 0.00** | **0.92 $\pm$ 0.02** | **0.67 $\pm$ 0.04** | **0.97 $\pm$ 0.02** | **0.88 $\pm$ 0.06** |
| NS$_{hom}$ | 0.95 $\pm$ 0.03 | 0.64 $\pm$ 0.06 | 0.86 $\pm$ 0.02 | 0.52 $\pm$ 0.03 | **1.00 $\pm$ 0.00** | **1.00 $\pm$ 0.00** | **0.92 $\pm$ 0.02** | 0.63 $\pm$ 0.05 | 0.95 $\pm$ 0.01 | 0.57 $\pm$ 0.06 |
| Z-score | 0.95 $\pm$ 0.01 | 0.57 $\pm$ 0.06 | 0.86 $\pm$ 0.03 | 0.55 $\pm$ 0.04 | **1.00 $\pm$ 0.00** | **1.00 $\pm$ 0.00** | 0.91 $\pm$ 0.02 | 0.57 $\pm$ 0.05 | 0.82 $\pm$ 0.04 | 0.53 $\pm$ 0.09 |
| Altman | 0.95 $\pm$ 0.01 | 0.48 $\pm$ 0.05 | 0.87 $\pm$ 0.03 | 0.58 $\pm$ 0.06 | 0.99 $\pm$ 0.01 | 0.96 $\pm$ 0.03 | 0.91 $\pm$ 0.02 | 0.61 $\pm$ 0.03 | 0.81 $\pm$ 0.05 | 0.59 $\pm$ 0.11 |
| QCAD | 0.90 $\pm$ 0.01 | 0.28 $\pm$ 0.04 | **0.93 $\pm$ 0.02** | **0.64 $\pm$ 0.04** | 0.98 $\pm$ 0.01 | 0.96 $\pm$ 0.02 | 0.86 $\pm$ 0.01 | 0.45 $\pm$ 0.07 | 0.96 $\pm$ 0.03 | 0.85 $\pm$ 0.11 |
| ROCOD | 0.93 $\pm$ 0.01 | 0.40 $\pm$ 0.05 | 0.79 $\pm$ 0.02 | 0.34 $\pm$ 0.05 | 0.90 $\pm$ 0.04 | 0.79 $\pm$ 0.08 | 0.84 $\pm$ 0.03 | 0.46 $\pm$ 0.07 | 0.78 $\pm$ 0.03 | 0.31 $\pm$ 0.05 |
| IForest | 0.77 $\pm$ 0.01 | 0.05 $\pm$ 0.01 | 0.62 $\pm$ 0.02 | 0.08 $\pm$ 0.02 | 0.83 $\pm$ 0.03 | 0.33 $\pm$ 0.06 | 0.71 $\pm$ 0.05 | 0.10 $\pm$ 0.02 | 0.79 $\pm$ 0.06 | 0.32 $\pm$ 0.10 |
| LOF | 0.92 $\pm$ 0.00 | 0.39 $\pm$ 0.03 | 0.49 $\pm$ 0.06 | 0.06 $\pm$ 0.01 | 0.92 $\pm$ 0.02 | 0.78 $\pm$ 0.04 | 0.61 $\pm$ 0.05 | 0.08 $\pm$ 0.02 | 0.71 $\pm$ 0.04 | 0.20 $\pm$ 0.04 |
| HBOS | 0.67 $\pm$ 0.01 | 0.04 $\pm$ 0.00 | 0.74 $\pm$ 0.05 | 0.27 $\pm$ 0.05 | 0.67 $\pm$ 0.03 | 0.26 $\pm$ 0.04 | 0.69 $\pm$ 0.04 | 0.10 $\pm$ 0.02 | 0.76 $\pm$ 0.05 | 0.34 $\pm$ 0.08 |

**Assessing the reliability of HDI** We evaluate using NS in a learning with abstention setting, where a model only offers a prediction on the test examples for which it is most confident. By abstaining when the model is uncertain, its performance on those examples for which it does offer a prediction should improve. We consider two different approaches for abstention when using NS. First, we use the HDI length $i(x,y)$: we sort all test points by their HDI length and abstain on the 5% with the widest HDI. Second, we fit an Isolation Forest on the contextual variables and abstain on the test points that have the top 5% most "outlying" contexts. Table 3 reports the ROC AUC and PR AUC computed over the 95% most confident examples identified by each approach. Across datasets the HDI-based abstention strategy yields superior or equal performance compared to the contextual baseline, supporting the use of $i(x,y)$ as a reliable indicator of epistemic uncertainty.

**RQ2: How sensitive is the framework to the choice of kernel and number of inducing points?**

To assess the robustness of the NS model, we evaluated the impact of different kernel functions and numbers of inducing points on predictive performance. Table 4 summarizes the variations in average expected NS and $i(x,y)$ when switching among kernel choices Rational Quadratic (RQ), Matérn 5/2, and RBF. The model proves stable across these kernels, showing only minor differences in performance, which confirms that the method does not rely heavily on a specific kernel assumption.

Table 5 illustrates the effect of increasing the number of inducing points from 5% to 20% of the training data. We observe that even with a small number of inducing points (5%), the model maintains reliable estimates of both mean and uncertainty, with only marginal benefits from further increases. This highlights a favorable trade-off between computational efficiency and predictive robustness, as the 5% configuration already achieves strong and consistent performance across datasets. However, training time increases significantly with more inducing points, scaling from approximately 10 minutes (5%) to around 70 minutes (20%) on an NVIDIA A6000 GPU.

**RQ3: Does $i(x,y)$ correlate with sparsity in contextual feature space?**

We expect EU to concentrate in underrepresented (sparse) regions in the contextual space since the model has fewer training examples to rely on. In this experiment, we aim to verify that $i(x,y)$ does indeed align with sparse contextual regions. Specifically, we evaluate the ordinal association between $i(x,y)$ with the anomaly scores $s(x)$, computed on contextual variables only. Here $s(x)$ is obtained by applying three classic anomaly detectors: IForest, LOF and HBOS. The association is quantified by the Weighted Kendall Tau correlation coefficient (Shieh, 1998). This rank-based metric is particularly suitable for comparing continuous variables, as it accounts for the magnitude of rank differences by assigning weights to the comparisons. In this version of the Kendall Tau coefficient, greater importance is given to elements higher in the rank. The coefficient is defined as:

$$\tau_w = \frac{\sum_{i<j} w_{ij} \cdot \text{sgn}[(x_i - x_j) \cdot (y_i - y_j)]}{\sum_{i<j} w_{ij}},$$

where $x_i$ and $x_j$ are the values of one variable, $y_i$ and $y_j$ are the values of the other variable, and sgn denotes the sign function. Weights $w_{ij}$ are used to emphasize certain pairs $(i,j)$ based on their significance

Table 3: ROC and PR AUC when making predictions on the 95% most certain test instances as determined by i(x,y) and IForest on contextual space when using the NS.

| Dataset | Abstain using $i(x,y)$ | | Abstain using IForest | |
|---|---|---|---|---|
| | ROC AUC | PR AUC | ROC AUC | PR AUC |
| Abalone | **0.97** | **0.71** | 0.96 | 0.66 |
| Concrete | **0.92** | **0.65** | 0.86 | 0.55 |
| SynMachine | **1.00** | **1.00** | **1.00** | **1.00** |
| Toxicity | **0.95** | **0.74** | 0.92 | 0.70 |
| Yacht | **1.00** | **0.95** | 0.97 | 0.87 |

Table 4: Impact of kernel choice on NS statistics.

| Kernel comparison | $\Delta\mathbb{E}[\text{NS}]$ | $\Delta i(x,y)$ |
|---|---|---|
| Matérn 5/2 vs RQ | 0.07 | 0.02 |
| RBF vs RQ | 0.04 | 0.01 |

Table 5: Effect of inducing points ratio on NS and train time.

| Inducing pts ratio | $\Delta\mathbb{E}[\text{NS}]$ (with 5%) | train time |
|---|---|---|
| 5% | – | $\sim$10 min |
| 10% | 0.06 | $\sim$40 min |
| 20% | 0.10 | $\sim$70 min |

Table 6: Weighted Kendall Tau correlation between $i(x,y)$ and anomaly scores $s(x)$.

| Dataset | IForest | LOF | HBOS |
|---|---|---|---|
| Abalone | 0.71 | 0.66 | 0.62 |
| Concrete | 0.66 | 0.66 | 0.64 |
| SynMachine | 0.65 | 0.62 | 0.60 |
| Toxicity | 0.73 | 0.68 | 0.63 |
| Yacht | 0.68 | 0.70 | 0.65 |

or importance in the dataset, often assigning larger weights to pairs involving elements with higher ranks. The results in Table 6 show a consistent positive correlation, confirming that that $i(x,y)$ can effectively capture sparsity in the contextual feature space.

### 4.4 Results on the Real-world Medical Data

**RQ4: Can NS aid in clinical practice for assessing the normalcy of the aortic diameter?**

For this experiment we train on normal subjects and test on adult patients with a diagnosis of bicuspid aortic valve, a condition typically associated with aorta dilations. We use the 40mm dilation threshold defined by current medical guidelines (Isselbacher et al., 2022) to assign patients to the positive class. This threshold is used solely to determine the ground truth labels needed for the ROC analysis and qualitative interpretation. NS assesses normalcy as a continuous quantity conditioned on patient context and is not trained on or tuned to this 40 mm cut-off.

Figure 2 shows the ROC for detecting anomalous SoV and AA aortic diameters, respectively. In both cases, our proposed approach results in a superior ROC curve and higher ROC AUC values (reported in the figure's legend) than its contextual competitors. To verify the significance of the difference between the ROC curves, the Delong test (DeLong et al., 1988) was applied, revealing statistically significant differences with QCAD and ROCOD ($p < 0.001$) for both SoV and AA.

One of the key advantages of NS is its ability to quantify EU via $i(x,y)$. Heatmaps in Figure 3 visualize $i(x,y)$ (males and females separately) at a fixed diameter of 32 mm (SoV and AA). To reduce the number

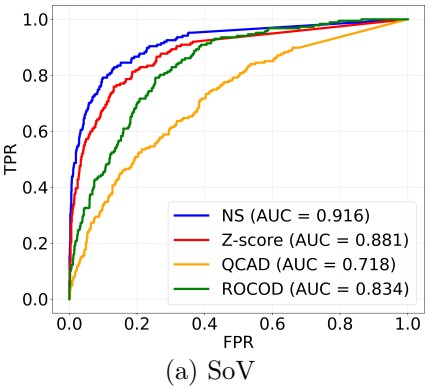 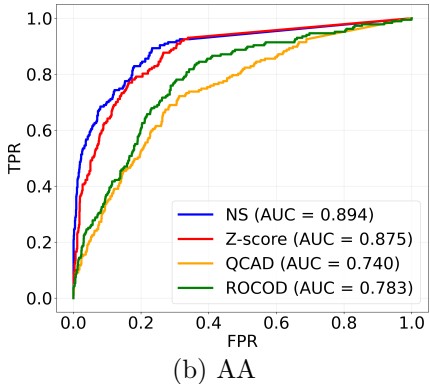

(a) SoV          (b) AA

Figure 2: ROC curves for detecting aortic dilation for SoV and AA diameters.

of variables and enable 2D visualization, we use the body surface area (BSA, proportional to $\sqrt{W \times H}$) in the plots. Dark red regions indicate subpopulations with greater EU and correspond to regions of sparsity in the contextual space. This information can help clinicians identify and flag patients for whom the model's predictions are less reliable, thus prioritizing additional diagnostic tests or closer monitoring. Such insights are especially valuable in clinical decision-making, where accurate and interpretable risk assessments are critical.

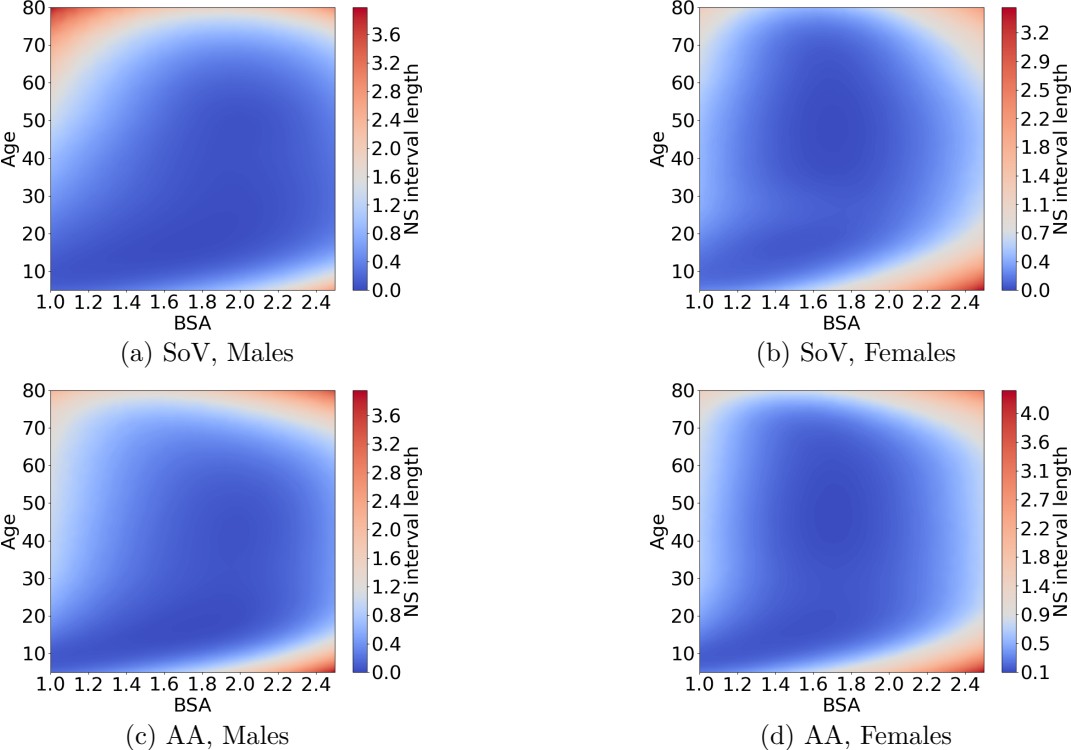

(a) SoV, Males          (b) SoV, Females

(c) AA, Males          (d) AA, Females

Figure 3: Heatmaps of $i((\text{Age}, \text{BSA}), y)$ at a fixed diameter $y = 32$mm for SoV and AA diameters.

Figure 4 highlights four representative cases where the linear Z-score predicts an anomaly score below the clinical threshold $Z = 2$ for at least one of the two aortic diameters (SoV or AA). However, the NS interval reveals significant diagnostic uncertainty, indicating that these patients may have a non-negligible probability of exceeding $Z = 2$.

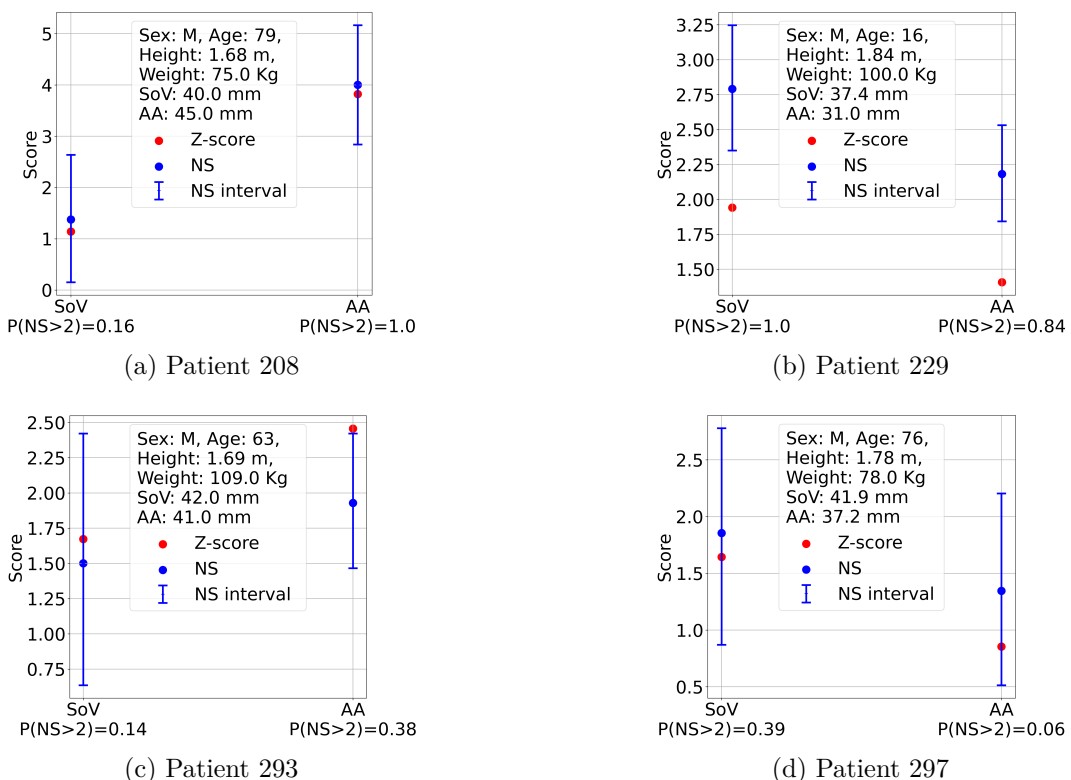

Figure 4: Bicuspid patients where NS interval highlights high diagnostic uncertainty. For each diameter, we show the Z-score (red), NS (blue), its HDI (error bar), and the corresponding probability $P(\text{NS} > 2)$.

In clinical practice, it may also be intuitive to report the posterior probability that NS exceeds a clinically meaningful threshold, rather than inspecting the full high-density interval. For the bicuspid cohort, we therefore compute the exceedance probability $P(\text{NS}(x, y) > 2)$ for each patient and report it alongside the per-diameter intervals in Figure 4. This summary can help flag borderline cases beyond point estimates: among patients with $\mathbb{E}[\text{NS}] < 2$, 82 (SoV) and 48 (AA) still have $P(\text{NS} > 2) > 0$ (i.e., a non-zero probability of crossing the anomaly threshold).

# 5    Conclusion

In this paper, we introduced *normalcy score* (NS), a framework for contextual anomaly detection that explicitly models both aleatoric and epistemic uncertainty. By leveraging heteroscedastic Gaussian process regression, our approach goes beyond traditional Z-scores by treating the anomaly score as a random variable. This enables the computation of high-density intervals, offering a more robust and interpretable assessment of anomalies.

Through experiments on benchmark datasets and a real-world application in cardiology, we demonstrated that NS achieves state-of-the-art performance comparing with other contextual anomaly detection methods such as QCAD and ROCOD. Additionally, the ability to quantify uncertainty provides significant advantages in high-stakes domains like healthcare, where decision-making must often account for sparse or unreliable data. Developing approaches that explicitly reason about predictive uncertainty is important for promoting trust in machine-learning systems. This is particularly true in high-stakes applications such as medicine, where predictions may influence decision-making and it is paramount to know when an output should be treated with caution or further investigated. Although NS sometimes offers only a modest margin over a plain Z-score, our experiments are, to our knowledge, the first to reveal that such a simple baseline can already

be remarkably competitive on contextual anomaly detection tasks, an observation that future benchmarks should not overlook.

Future work will explore extending the NS framework to other domains where contextual information plays a key role, like in the aortic diameters scenario. In addition, the case of vector-valued behavior needs to be addressed by taking into account the relationships among different behavioral variables. For example, the overall shape of the aorta might be anomalous even though every individual diameter (such as SoV or AA) may be normal. This can be addressed with multi-output/multi-task Gaussian processes (Bonilla et al., 2007), and more structured constructions that place Wishart process priors over context-dependent covariance matrices to explicitly capture correlations between behavioral variables (Heaukulani & van der Wilk, 2019).

Overall, NS represents a significant step forward in contextual anomaly detection, providing a robust, interpretable, and uncertainty-aware solution to this challenging problem.

### Acknowledgments

JD acknowledges the support of the Flemish government under the "Onderzoeksprogramma Artificiële Intelligentie (AI) Vlaanderen" programme. A special thanks to Curzio Checcucci for his valuable feedback during the initial formulation of the problem.

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
