# OpenReview forum: "Dealing with Uncertainty in Contextual Anomaly Detection"
_TMLR — Accepted by TMLR_

### Review · Reviewer_mKdL · 2025-10-14

**Summary Of Contributions:**

The paper proposes a Gaussian Process Regression (GPR) based framework, Normalcy Score (NS), for the Contextual Anomaly Detection (CAD) task, predicting not only the level of anomaly but also the level of uncertainty for the prediction. Instead of outputting a single anomaly score, such as z-score, the framework regards it as a random variable and outputs a distribution of it, allowing computation of a confidence interval (HDI). It also explicitly considers and disentangles two sources of uncertainty: aleatoric uncertainty (AU) from the nature of the task, and epistemic uncertainty (EU) from the lack of datapoints. In practice, this could greatly improve the interpretability of the predictions by flagging which predictions are not as confident, as well as how large the confidence interval is.

Strengths
1. Quantitatively modeling confidence level of predictions is conceivably helpful in practice.
2. The proposed framework is mathematically solid and neat.
3. The paper is well written, making the logic rather easy to follow.

Weaknesses
1. Lack enough analyses of the reliability and superiority of the estimated confidence level (HDI), one of the major contributions of the paper. Some analyses could be:
- Does the 95% HDI truly cover the real z-score calculated or approximated from real data about 95% of the time?
- Does HDI-based confidence estimation outperform other competitive approaches (or simple baselines such as sample density thresholding in feature space)?
2. Regarding interpretability, HDI is logically interpretable but not necessarily practically. For instance, the probability of a data point being anomalous (e.g. probability of having a z-value > 2.0) may be practically more straightforward, and could actually be calculated from the already-predicted distribution of z-score.

**Audience:**

Yes

**Audience Explanation:**

The proposed framework that predicts anomaly along with a confidence score (HDI) improves prediction interpretability, which is conceivably helpful in practice.

**Claims And Evidence:**

Yes

**Claims Explanation:**

Despite lack of enough analysis and comparisons for the proposed HDI, the proposed framework is mathematically solid and neat. Furthermore, some sanity check analysis are conducted to show the alignment between model predictions and human interpretation, e.g. high correlation between prediction confidence and data point density in the feature space.

**Requested Changes:**

The following are recommendations for further strengthening the work, rather than requests.
1. Include some analyses of the reliability and superiority of the estimated confidence level (HDI). This could help link model predictions to real world data, and establish a solid baseline for the path that future works can compare against. Some candidate analyses include:
- Does the 95% HDI truly cover the real z-score calculated or approximated from real data about 95% of the time?
- Does HDI-based confidence estimation outperform other competitive approaches (or simple baselines such as sample density thresholding in feature space)?
2. Propose some variables that can be quickly derived from the predicted distribution of z-scores that are practically more straightforward than HDI, such as the probability of a data point being anomalous (e.g. probability of having a z-value > 2.0).

---

> ### Author Response · Authors · 2025-12-15
>
> We thank the reviewer for these thoughtful and constructive recommendations, which we believe will further strengthen the practical relevance of the paper. Below we respond to each point and describe the corresponding changes we will incorporate in the revised manuscript.
>
> ---
>
> > Include some analyses of the reliability and superiority of the estimated confidence level (HDI). This could help link model predictions to real world data, and establish a solid baseline for the path that future works can compare against
>
>
> We thank the reviewer for this helpful suggestion. To directly compare HDI-based confidence filtering against a simple feature-space heuristic, we extend the abstention/filtering experiment in Table 3 as follows. In addition to discarding test instances with large epistemic uncertainty (i.e., removing the top 5% points by HDI length $i(x,y)$), we also consider a purely contextual baseline that removes points in sparse regions of the contextual space by fitting an Isolation Forest on contextual variables only and discarding the top 5% most “outlying” contexts. We then evaluate on the remaining test instances. We will include this additional comparison in the revised manuscript and discuss it alongside the HDI-based filtering results.
>
> **NS performance (ROC AUC / PR AUC) after removing the top 5% most uncertain/sparse test instances**
>
> | Dataset    | Filter by $i(x,y)$ (top 5% removed) | Filter by contextual IForest score (top 5% removed) |
> |------------|-----------------------------------------------|-----------------------------------------------------|
> | Abalone    | 0.97 / 0.71                                     |       0.96 / 0.66                                    |
> | Concrete   | 0.92 / 0.65                                     |       0.86 / 0.55                                   |
> | SynMachine | 1.00 / 1.00                                     |     1.00 / 1.00                                      |
> | Toxicity   | 0.95 / 0.74                                     |   0.92 / 0.70                                       |
> | Yacht      | 1.00 / 0.95                                     |           0.97 / 0.87                                  |
>
> ---
>
> > Propose some variables that can be quickly derived from the predicted distribution of z-scores that are practically more straightforward than HDI, such as the probability of a data point being anomalous (e.g. probability of having a z-value > 2.0).
>
> We thank the reviewer for this excellent suggestion. We agree that simple quantities derived from the NS posterior can be very useful in practice, especially in our medical application where clinicians are familiar with threshold-based reasoning (e.g., “Z-score > 2”). In particular, reporting the posterior probability of exceeding a clinically meaningful threshold, such as $P(\mathrm{NS}(x,y)>2)$, provides an immediate and interpretable summary of risk that complements both the point estimate and the HDI.
>
> To explore this direction, we ran an additional analysis on the bicuspid cohort (used as the test set) and focused on patients whose expected score satisfies $\mathbb{E}[\mathrm{NS}] < 2$, but for whom the posterior still assigns non-zero mass above the threshold. The table below reports how many such patients have a non-negligible exceedance probability at different levels, separately for SoV and AA.
>
> **Bicuspid test cohort: patients with $\mathbb{E}[\mathrm{NS}] < 2$ but $P(\mathrm{NS}>2)$ above a threshold**
>
> | Threshold on P(NS>2) | SoV (count) | AA (count) |
> |------------------------|------------|-----------|
> | > 0                    |          82 |        48 |
> | > 0.10                 |         18 |        14 |
> | > 0.20                 |         12 |         7 |
> | > 0.30                 |          9 |         5 |
>
>
>
> As shown by this analysis there is a non-trivial subset of patients whose expected score is below 2 while the posterior assigns a non-negligible probability to exceeding the threshold. This kind of summary can therefore flag borderline cases where the point estimate alone may appear reassuring, but uncertainty-aware assessment suggests caution. We thank the reviewer again for motivating this analysis, and we will incorporate it in the revised manuscript.

---

### Review · Reviewer_KHaV · 2025-10-28

**Summary Of Contributions:**

This paper introduces a novel method for contextual anomaly detection.
The method uses two Gaussian Processes to model mean and log standard deviation of the target variables conditioned on the context.
The proposed methods can be used to obtain point estimates and high-density intervals.

**Audience:**

Yes

**Audience Explanation:**

The proposed method is novel, straight-forward, and effective.

**Broader Impact Concerns:**

None.

**Claims And Evidence:**

No

**Claims Explanation:**

The experiments are limited to five small datasets and six shallow baselines. The proposed method only shows slight improvements over the next best baselines on three out of five datasets. Claims of consistent and superior performance would be much more convincing with more baselines, in particular deep baselines, and more datasets.

**Requested Changes:**

The claims of the paper would be much stronger with extended experiments including more recent baselines and datasets.

---

> ### Author Response · Authors · 2025-11-03
>
> We thank the reviewer for the careful reading and constructive feedback. Below we clarify our experimental choices, explain key design decisions, and outline the edits planned for the revision.
>
> **Datasets, baselines, and interpretability (CAD comparability).**
> Our evaluation mirrors the de-facto CAD setup used by the sota contextual anomaly detection methods ROCOD (*Liang et al.*) and QCAD (*Li et al.*). We selected benchmark datasets from previous CAD works that are compatible with our setup (i.e., datasets with explicit contextual variables and a scalar behavioral variable) so that results are strictly comparable. This single variable focus is deliberate and tied to interpretability: in our scenario anomaly score is given by the deviation of the behavioral variable from its context-specific expectation.
> A multi-task extension is conceivable as future work and would likely be preferable to the aggregation across behavioral variables adopted in prior work (as stated in Conclusion).
> In addition to these benchmarks, we also included a real medical dataset on the aorta, which highlights the practical value of modeling normalcy and the clarity of our explanations for end-users.
>
> **Non-contextual AD baselines and why they underperform.**
> For non-contextual AD, we used the standard baselines commonly reported in CAD studies (e.g., IF/LOF/HBOS). As also observed in prior work, these methods perform substantially below CAD approaches not because of the particular anomaly detector per se, but because they do not separate contextual covariates from the behavioral variable (i.e., they do not explicitly model $p(y \mid x)$ but $p(y, x)$). This is precisely the point illustrated by our cartoon example (see Fig. 1). In contrast, CAD methods such as ROCOD and QCAD condition on context and thereby define anomalies as deviations of the behavioral variable given the context.
> We will remark this point in the paper by adding a concise paragraph stating that the observed gap stems from the lack of explicit contextual modeling, rather than from which traditional detector is chosen.
>
> **Performance and the strength of the Z-score.**
> We agree that a “simple” Z-score can be highly competitive in CAD. A contribution of our paper is also to show how strong the Z-score actually is under CAD evaluation, on some datasets it matches or surpasses sophisticated CAD techniques, a point that, to the best of our knowledge, has not been included in prior CAD studies. We will make this explicit in the revision, clarifying that the Z-score baseline has often been omitted or under-emphasized and documenting its competitiveness in the CAD setting.
>
> In direct response to the reviewer’s concern about the breadth of baselines, we expanded our benchmark comparison with two additional methods that sit between the linear Z-score and our full heteroscedastic GP model: a homoscedastic GP variant, reported as NS_homoscedastic; and a linear variance-estimation baseline  (Altman method).
>
> **Metrics: ROC AUC / PR AUC (mean ± std)**
>
> | Dataset    | NS | Z-score | NS_homoscedastic | Altman |
> |------------|--------------------------|-------------------------|------------------------------|----------------------------|
> | Abalone    | 0.96 ± 0.01 / 0.65 ± 0.04 | 0.95 ± 0.01 / 0.57 ± 0.06 | 0.95 ± 0.03 / 0.64 ± 0.06 | 0.95 ± 0.01 / 0.48 ± 0.05 |
> | Concrete   | 0.89 ± 0.02 / 0.60 ± 0.01 | 0.86 ± 0.03 / 0.55 ± 0.04 | 0.86 ± 0.02 / 0.52 ± 0.03 | 0.87 ± 0.03 / 0.58 ± 0.06 |
> | SynMachine | 1.00 ± 0.00 / 1.00 ± 0.00 | 1.00 ± 0.00 / 1.00 ± 0.00 | 1.00 ± 0.00 / 1.00 ± 0.00 | 0.99 ± 0.01 / 0.96 ± 0.03 |
> | Toxicity   | 0.92 ± 0.02 / 0.67 ± 0.04 | 0.91 ± 0.02 / 0.57 ± 0.05 | 0.92 ± 0.02 / 0.63 ± 0.05 | 0.91 ± 0.02 / 0.61 ± 0.03 |
> | Yacht      | 0.97 ± 0.02 / 0.88 ± 0.06 | 0.82 ± 0.04 / 0.53 ± 0.09 | 0.95 ± 0.01 / 0.57 ± 0.06 | 0.81 ± 0.05 / 0.59 ± 0.11 |
>
> In the revised manuscript we will include this expanded comparison and discuss the results.

---

### Review · Reviewer_DdVc · 2025-12-07

**Summary Of Contributions:**

- The authors propose the use of heteroscedastic gaussian process regression as a model for contextual anomaly detection- a setting where characterization of anomalies in so called "behavioral variables" depends upon the value of simultaneously observed "contextual variables".
- After fitting gaussian process models to data using a variational approximation based on inducing points, the authors define various statistics from the heteroscedastic GP regression fits which can be used to detect anomalous samples, built around a normalcy score (NS): a distributional extension of traditional Z scoring.
- With these statistics in hand, they evaluate the performance of NS based point estimates in detecting contextual anomalies, compared to other methods formulated specifically for contextul anomaly detection; e.g. Quantile Contextual Anomaly Detection (QCAD), Robust Contextual Outlier Detection (ROCOD), and a simple Z scoring baseline, as well as non-contextual methods for outlier detection such as isolation forests. They show that NS compares favorably to other methods in a large majority of cases in various datasets into which anomalies are artificially injected through perturbations of real observations.
- The authors then go on to characterize the variance in the normalcy score distribution, showing that it appears to widen (reflecting more epistemic uncertainty) in locations where data are sparse.
- Finally, they apply their normalcy score based analyses to real-world medical data, in which anomalies in aortic diameter were analyzed using normalcy score based metrics, as well as other competing methods. Normalcy score was found to perform well in detecting anomalies against human defined ground truth, and likewise was used to characterize model prediction confidence as a function of various contextual variables.


A major strength of the method proposed by the authors is interesting in that it appears conceptually more straightforwards and effective than many previously proposed methods in the field.

A major weakness of the paper is lack of clarity. It is difficult to evaluate claims made about the model's ability to separate different sources of uncertainty as the principal driver of the model's success.

**Audience:**

Yes

**Audience Explanation:**

The method presented here appears to be at least as effective as previously discussed methods for contextual anomaly detection in a wide variety of tasks, while presenting a Bayesian approach which is conceptually more straightfowards than existing proposals for this problem. I believe these qualities would make this work interesting for the TMLR audience.

**Broader Impact Concerns:**

Uncertainty estimation is often framed with applications towards high-risk settings, such as clinical practice as described in this work. Considering the ethical implications of trusting the system under consideration with such decisions would be useful to describe in a broader impact statement.

**Claims And Evidence:**

No

**Claims Explanation:**

- As stated above, I found that lack of clarity made it difficult to evaluate the actual details of the proposed method, and alignment with the claims made, especially around the importance of separating epistemic and aleatoric uncertainty. I will group together specific examples related to this point in the requested changes section below.
- I attempted to locate appendices where experimental details might be explained in greater detail, but I was unable to find them. Please let me know if I have missed something.

**Requested Changes:**

### Methodological details.
The following details should be specified somewhere in the main manuscript (or clearly referenced into an appendix), as I believe they are important to evaluate the claims made.
- **Implementation details of QCAD.** The default implementation of QCAD from Li and van Leeuwen 2022 describes hyperparameter settings which reflect a balance of logistical constraints and effective performance (e.g. number of trees = 10, maximal features = total features). Were these parameters explored in your comparison? I believe this question to be important because performance differenes between NS and QCAD are quite small in a variety of cases.
- **Z scoring baseline.** My understanding of this model is that it represents the z score estimated by fitting a linear regression model from contextual variables to behavioral variables. It would be useful to explicitly define this model, as claims are made as the surprising effectiveness of this simple baseline.
- **Fitting paradigm.** Please describe in greater detail the fitting procedures used to infer gaussian processes $f_1,f_2$ in each experiment, as well as corresponding inference procedures for other methods. In particular:
	- It is not clear from the main text that $f_1,f_2$ are fit jointly as a single, heteroscedastic gaussian process, and that the parameters of $f_1,f_2$ are thus linked through inference of a shared posterior process. Please make this explicit in your description of the inference procedure.
	- For the UCI datasets, are anomalies injected into both the training and test set, or only the test set?
	- Relatedly, how are methods like QCAD, which do not explicitly separate out a training and test set, applied to the aorta diameter dataset?
	- In the description of the medical dataset, it is stated that `We use the 40mm dilation threshold defined by
current medical guidelines (Isselbacher et al., 2022) to assign patients to the positive class.` If I understand correctly, this task on its own would be an example of non-contextual anomaly detection, as the human defined threshold is not dependent upon any contextual variables. In general, more detail about the specific goals of the task, the available contextual variables, and the provided examples are necessary to make the value of NS in the example application clear. How should we interpret the example patients in Figure 4, where NS and Z scores give different anomaly scores? Does the fact that we are given the ground truth SoV and AA diameters for each patient provide a ground truth against which we should assess the anomaly scores? Or rather, are these simply illustrative examples of differences in model comparison?
	- Please comment on differences reported in scores from Table 6, Li and van Leeuwen and Table 2 in this paper, some of which are quite significant.
### Understanding the effectiveness of NS
The introduction to this paper closely relates its contributions to the notion of disentangling epistemic and aleatoric uncertainty. This connection is references a few times throughout the paper, but I found the specifics of this connection to be somewhat unclear. I have organized my questions below:

**Disentangling AU and EU.** Section 3.1 states that AU is captured by the second GP $f_2$, while EU is captured by the posterior of both $f_1,f_2$. Can we be more precise here? In particular:
- I would expect that the posterior *mean* of $f_2$ might not shrink with more data (reflecting AU) but that the posterior *variance* of $f_1$ and $f_2$ would (reflecting EU). If this is what is intended, it would be useful to provide a mathematical statement for precision. As much seems to be stated in section 3.2 ("`As a result, p(NS(x, y)) is not normal. Still, we wish to summarize it in terms of position and dispersion,
that quantify AU and EU, respectively.`") but at the moment these statements appear to be in conflict.
- The introduction states that anomalous points are exactly those with large AU. Thus, it would make sense to choose an anomaly metric which captures only aleatoric uncertainty. While $s(x,y)$ as defined in the paper seems very effective, wouldnt it be more appropriate to consider only the mean of $f_2$, instead of including the posterior variance variance in the provided pointwise estimate? Stated differently, I would expect $s(x,y)$ to shrink as we add more training data, which seems inappropriate for an estimate of AU.

**Important baseline models**. As noted by the authors, it is significant that a simple (linear?) z scoring baseline appears to perform very well in contextual anomaly detection tasks. This results suggests that at least some part of the effectiveness of NS may be shared with other methods which employ **any** kind of Z-scoring based formulation, as opposed to other methods which attempt estimation of contextual quantiles, or contextual neighborhoods (e.g. QCAD). It would be useful to consider how NS compares to the following simple baselines to understand how important the separation of EU and AU is to the overall success of this model.
- **Homoscedastic GP.** If one was to fit a simple homoskedastic GP to training data, and perform hyperparameter search on the marginal standard deviation, one could estimate z scores on test data similarly to how one does in a linear model, computing a residual relative to the mean function. Understanding how this model performs in anomaly detection would be useful as a model of intermediate complexity between NS and Z scoring.
- **Altman model.** As noted by the authors, Altman (1993) proposed variance estimation within a linear regression framework. Given the success of the simple linear Z score model, it would be useful to consider a simple alternative to NS anomaly detection based on this model. How well would such a model do without the nonparametric expressiveness of GP regression?

### Differences in problem setting
It would be useful to discuss differences in problem setting between this model and others which have been discussed for contextual anomaly detection.
- **Categorical contextual variables**. Are categorical contextual variables considered in the study? If not, it would be good to include this in the discussion, as a difference in problem setting to other methods which are based on trees or other models which work well with tabular data.
- **Multiple behavioral variables**. Other contextual anomaly detection methods consider multidimensional outputs, and devote a significant amount of attention to the question of how to attribute anomaly scores to different behavioral variables. While vector-valued behavior is discussed briefly at the end of the work, it should be made clear that this concern is a major part of the problem formulation in other studies of contextual anomaly detection.

### Others.

I believe the following requests would be useful for interested readers of the paper, but are not critical to securing my recommendation.
- An alternative method of building heteroscedastic Gaussian Processes which generalize well to higher dimensional output could be through the use of Wishart processes or inverse Wishart Processes, for which there are also efficient variational inference schemes (Heaukulani and van der Wilk, 2019).
- Another example of useful decompositions of uncertainty into epistemic and aleatoric components comes from Depeweg et al. 2018, which may be a useful reference.
- bottom of page 9: `Dark red regions indicate subpopulations with greater AU and correspond to regions of sparsity
in the contextual space.`: shouldn't this be EU?

## References
- Heaukulani, Creighton, and Mark van der Wilk. "Scalable Bayesian dynamic covariance modeling with variational Wishart and inverse Wishart processes." Advances in Neural Information Processing Systems 32 (2019).
- Depeweg, S., Hernandez-Lobato, J. M., Doshi-Velez, F., & Udluft, S. (2018, July). Decomposition of uncertainty in Bayesian deep learning for efficient and risk-sensitive learning. In International conference on machine learning (pp. 1184-1193). PMLR.

---

> ### Author Response · Authors · 2025-12-14
>
> We thank the reviewer for their careful reading of our manuscript and their constructive feedback. Below we respond point-by-point and describe the corresponding changes we will make in the revised version of the paper.
>
> ---
>
> ## 1. Methodological details
>
> > **Implementation details of QCAD.**
> >  The default implementation of QCAD from Li and van Leeuwen 2022 describes hyperparameter settings which reflect a balance of logistical constraints and effective performance (e.g. number of trees = 10, maximal features = total features). Were these parameters explored in your comparison?
>
>
> We thank the reviewer for raising this point. In all our experiments we used the official QCAD implementation and did not change its hyperparameters. We adopted the default settings from QCAD’s code, including:
>
> - the default number of trees in the quantile regression forest,
> - the default choice of maximal features per tree,
> - the default minimum node size,
> - the default number of conditional quantiles,
>
> In the revised manuscript we will emphasize this choice of default hyperparameters in the experimental setup.
>
> ---
>
> > **Z-scoring baseline.**
> > My understanding of this model is that it represents the z score estimated by fitting a linear regression model from contextual variables to behavioral variables. It would be useful to explicitly define this model, as claims are made as the surprising effectiveness of this simple baseline.
>
> We agree that this baseline should be stated more clearly. In our experiments, the Z-score baseline is exactly the Z-score defined in Section 2, with $f(x)$ instantiated as a linear regression fitted on the training data. The residual standard deviation is treated as a single global scalar, so the model is homoscedastic. In the revised manuscript, we will explicitly mention in the experimental section that our Z-score baseline corresponds to this linear, homoscedastic instantiation.
>
>
> ---
>
> > **Fitting paradigm.**
> > It is not clear from the main text that $f_1$, $f_2$ are fit jointly as a single, heteroscedastic gaussian process, and that the parameters of $f_1$, $f_2$ are thus linked through inference of a shared posterior process.
>
> NS is based on a heteroscedastic Gaussian process regression model with two latent GPs.
> These two processes are learned jointly as a single GP model using the heteroscedastic framework implemented in GPflow.
>
> Concretely, for each dataset and fold:
>
> - We place independent GP priors on $f_1$ and $f_2$ with Rational Quadratic kernels.
> - We use a sparse variational approximation with inducing points; inducing inputs and hyperparameters are initialized from the training data and optimized jointly.
> - Training is performed using Natural Gradient Descent for the variational parameters combined with Adam for kernel and likelihood hyperparameters.
> - Both the mean function $m_1(x)$ and log-variance function $m_2(x)$, as well as their posterior variances, are therefore inferred together from the same data.
>
> In the revised manuscript we will explicitly specify these implementation details.
>
> ---
>
> > **UCI datasets: anomaly injection.**
> > For the UCI datasets, are anomalies injected into both the training and test sets, or only the test set?
>
> For the UCI datasets we follow the perturbation scheme introduced in previous works and described in our Section 4.
>
> Importantly, this injection is performed before any train/test split. We then apply 5-fold cross-validation, so each fold contains a small contamination of anomalous points. This reflects the intended unsupervised anomaly-detection scenario.
>
> We will clarify this order (“inject first, then split”) explicitly in the experimental-setup section.

---

> ### Author Response · Authors · 2025-12-14
>
> > **Medical dataset and 40 mm threshold.**
> >  In general, more detail about the specific goals of the task, the available contextual variables, and the provided examples are necessary to make the value of NS in the example application clear.
>
> We thank the reviewer for raising this important clarification. The medical experiment is not designed as a purely binary classification task; rather, NS is used to assess normalcy as a continuous quantity for aortic diameters conditioned on patient context, and to quantify the model’s uncertainty. The 40 mm threshold from clinical guidelines is used as an external reference, essentially a sanity check for ROC analysis and qualitative interpretation, rather than as a contextual decision rule built into the model. In this setting, the contextual variables are age, sex, weight, and height (or body-surface area derived from them), while the behavioral variables are the aortic diameters measured at two locations, sinus of Valsalva (SoV) and ascending aorta (AA).
>
> The patients shown in Figure 4 are illustrative examples highlighting cases where a simple Z-score might be just below the classical anomaly threshold (e.g., $|Z| < 2$), but the NS high-density interval reveals substantial epistemic uncertainty about whether the measurement is actually normal for that context. We will extend the description of the medical experiment to emphasize these points and clarify that the per-patient examples are meant to illustrate qualitative differences between NS and Z-score.
>
> ---
>
> > **Differences with QCAD scores**
> > Please comment on differences reported in scores from Table 6, Li and van Leeuwen and Table 2 in this paper, some of which are quite significant.
>
> We thank the reviewer for pointing this out. The differences between the QCAD results reported in Li and van Leeuwen’s original paper and those in our Table 2 are generally minor (a few percentage points) across most datasets. The only substantial discrepancy arises on the Abalone dataset. In our setup, as specified in Table 1, we treat Rings as the sole behavioral variable and all remaining attributes as contextual, so Abalone is handled as a single-output (one behavioral variable) contextual anomaly detection problem. The deviation in scores for this specific dataset is therefore a direct consequence of this modelling choice, rather than of differences in the QCAD implementation itself.

---

> ### Author Response · Authors · 2025-12-14
>
> ## 2. Understanding the effectiveness of NS
>
> > **Disentangling AU and EU.**
> > I would expect that the posterior mean of $f_2$ might not shrink with more data (reflecting AU) but that the posterior variance of $f_1$ and $f_2$ would (reflecting EU). If this is what is intended, it would be useful to provide a mathematical statement for precision.
>
> This is indeed the case and we will provide more details, which we summarize here. In our model, $f_1(x)$ and $f_2(x)$ are independent GPs with posterior means $m_1(x)$, $m_2(x)$ and posterior variances $\sigma_1^2(x)$, $\sigma_2^2(x)$. As more data are observed in a neighbourhood of $x$, the variances $\sigma_1^2(x)$ and $\sigma_2^2(x)$ shrink, while the means converge to the underlying regression functions. Aleatoric uncertainty at a given context is represented by the input-dependent noise scale $\exp(m_2(x))$,
> which captures the intrinsic variability of $y \mid x$ and does not vanish even in the limit of infinite data. Epistemic uncertainty, by contrast, is encoded in the posterior variances $\sigma_1^2(x)$ and $\sigma_2^2(x)$, which decrease as the local sample size grows.
>
> The NS itself is a random variable $\mathrm{NS}(x,y) = (y - f_1(x))\,\exp(-f_2(x))$, which is similar in spirit to the traditional (pointwise) Z-score but whose distribution (which is non-Gaussian due to the log-normal term) reflects EU. In the paper we summarize this distribution by (i) its expectation $s(x,y)=\mathbb{E}[\mathrm{NS}(x,y)]$ and (ii) the length of a high-density interval $i(x,y)$, which is driven by $\sigma_1^2(x)$ and $\sigma_2^2(x)$ and therefore reflects epistemic uncertainty. As data accumulate near $x$, $i(x,y)$ shrinks. As suggested by another reviewer, we may as well introduce other summaries such as the probability that $\mathrm{NS}(x,y)$ is greater than a certain value (for example the typically used threshold $Z>2$ in studies related to aorta normalcy) and we will report this in the revised paper.
>
> Importantly, when computing the pointwise estimate $s(x,y)$, the expectation involves a multiplicative correction term that depends on $\sigma_2^2(x)$. This factor appears because $\exp(-f_2(x))$ is log-normally distributed under the GP posterior, and we use the exact closed-form expression of $\mathbb{E}[NS(x, y)]$ (rather than replacing $f_2(x)$ by its posterior mean). This correction term converges to $1$ as $\sigma_2^2(x)$ decreases, so it has a negligible effect in well-supported regions and primarily ensures that the point estimate remains consistent with the exact posterior expectation. In the revised version we will make this derivation and its interpretation explicit in Section 3 to avoid any apparent inconsistency with Section 3.2.
>
>
>
> ---
>
> > **Choice of anomaly metric.**
> > The introduction states that anomalous points are exactly those with large AU. Thus, it would make sense to choose an anomaly metric which captures only aleatoric uncertainty.
>
> We agree with the reviewer that, in principle, contextual anomalies are characterized by unusually large deviations relative to the context-dependent noise level, i.e., by the (contextual) aleatoric component encoded by the mean of $f_2(x)$. However, AU can be only exactly quantified in the limit of large data, when EU has vanished. This is why in our approach we distinguish between what constitutes an anomaly under $p(y\mid x)$ and (ii) how confident we are about the score assigned by the model at a given context. The former is governed by the noise scale implied by $m_2(x)$ (AU), while the latter depends on the posterior uncertainty in both $f_1$ and $f_2$ (EU), which is larger in underrepresented regions of the contextual space.
>
> This motivates our choice of complementing the expected $Z$ score with an uncertainty quantification: the posterior variances $\sigma_1^2(x)$ and $\sigma_2^2(x)$ are not used to redefine normalcy, but to assess the reliability of the score and enable abstention/flagging when the context is sparse. In the revised manuscript we will better clarify this distinction.

---

> ### Author Response · Authors · 2025-12-14
>
> > **Important baseline models (homoscedastic GP and Altman model).**
> > As noted by the authors, it is significant that a simple (linear?) Z-scoring baseline appears to perform very well in contextual anomaly detection tasks. This suggests that part of the effectiveness of NS may be shared with other Z-scoring based formulations. It would be useful to compare NS to: (i) a homoscedastic GP baseline, and (ii) the Altman (1993) variance-estimation model within a linear regression framework.
>
> We thank the reviewer for this insightful suggestion. When AU does not depend on $x$ and dataset are large, the distribution of our score becomes concentrated and in the limit of infinite data it would become equivalent to the traditional Z-score, but for heteroscedastic and finite datasets this is not the case and we need to resort to empirical evaluations. We thus implemented the suggested additional baselines: (i) the homoscedastic GP variant and (ii) the Altman (1993) baseline (which we re-implemented following the original paper, with explicit variance estimation from residuals).
>
> **Metrics: ROC AUC / PR AUC (mean ± std)**
>
> | Dataset    | NS | Z-score | NS_homoscedastic | Altman |
> |------------|--------------------------|-------------------------|--------|--------|
> | Abalone    | 0.96 ± 0.01 / 0.65 ± 0.04 | 0.95 ± 0.01 / 0.57 ± 0.06 | 0.95 ± 0.03 / 0.64 ± 0.06 | 0.95 ± 0.01 / 0.48 ± 0.05 |
> | Concrete   | 0.89 ± 0.02 / 0.60 ± 0.01 | 0.86 ± 0.03 / 0.55 ± 0.04 | 0.86 ± 0.02 / 0.52 ± 0.03 | 0.87 ± 0.03 / 0.58 ± 0.06 |
> | SynMachine | 1.00 ± 0.00 / 1.00 ± 0.00 | 1.00 ± 0.00 / 1.00 ± 0.00 | 1.00 ± 0.00 / 1.00 ± 0.00 | 0.99 ± 0.01 / 0.96 ± 0.03 |
> | Toxicity   | 0.92 ± 0.02 / 0.67 ± 0.04 | 0.91 ± 0.02 / 0.57 ± 0.05 | 0.92 ± 0.02 / 0.63 ± 0.05 | 0.91 ± 0.02 / 0.61 ± 0.03 |
> | Yacht      | 0.97 ± 0.02 / 0.88 ± 0.06 | 0.82 ± 0.04 / 0.53 ± 0.09 | 0.95 ± 0.01 / 0.57 ± 0.06 | 0.81 ± 0.05 / 0.59 ± 0.11 |
>
> In the revised manuscript we will include this comparison and discuss the results.
>
>
> ---
>
> ## 3. Differences in problem setting
>
> > **Categorical contextual variables.**
> > Are categorical contextual variables considered in the study? If not, it would be good to include this in the discussion, as a difference in problem setting to other methods which are based on trees or other models which work well with tabular data.
>
> Yes. Our study includes categorical contextual variables. For instance, the clinical dataset includes sex as a categorical contextual variable. Moreover, some of the UCI benchmarks also contain categorical attributes. In the revised manuscript we will make this explicit in the experimental section.
>
>
> ---
>
> > **Multiple behavioral variables.**
> > Other contextual anomaly detection methods consider multidimensional outputs, and devote a significant amount of attention to the question of how to attribute anomaly scores to different behavioral variables. While vector-valued behavior is discussed briefly at the end of the work, it should be made clear that this concern is a major part of the problem formulation in other studies of contextual anomaly detection.
>
>
> We agree that a principled multi-behavioral (multi-output) formulation is an important direction, and we will make clearer in the revised manuscript that this aspect is central in several strands of prior work on contextual anomaly detection. NS can naturally be extended to this setting using multi-output GP constructions or Wishart-process priors over covariance matrices, which would explicitly capture correlations between behavioral variables.
>
> At the same time, we will emphasize that in a large portion of prior CAD literature, multi-behavioral outputs are often handled in a largely heuristic way by computing anomaly scores per behavioral variable and then aggregating them (e.g., via averaging or summation), rather than through a fully principled joint probabilistic model of the behavioral vector. We will expand the discussion accordingly and position NS with respect to these existing multi-dimensional CAD approaches.
>
>
> ---
>
> ## 4. Other comments
>
> We thank the reviewer for these helpful suggestions. In the revised manuscript we will add the proposed references, including Heaukulani eet al. (2019) on Wishart/inverse-Wishart process constructions for heteroscedastic multi-output GPs, and Depeweg et al. (2018) as an additional perspective on decomposing epistemic and aleatoric uncertainty. We also thank the reviewer for spotting the typo at the bottom of page 9: the dark red regions correspond to EU, not AU, and we will correct this in the revision.

---

> > ### Comment · Reviewer_DdVc · 2025-12-15
> > **Thank you.**
> >
> > Thanks to the paper authors for their thoughtful response. If the revisions discussed above are integrated into the existing manuscript, I believe this would be sufficient to address my major concerns about claims made by the paper. I look forwards to seeing the updated manuscript.

---

### Author Response · Authors · 2025-12-19

We thank the reviewers for their time and constructive feedback, which helped us improve the manuscript. In the revised version, we addressed all comments and implemented several changes to enhance clarity, methodological transparency, and the presentation of the results. **All modifications are highlighted in blue in the revised manuscript.**

In particular, the main updates are as follows:

- **Methodological clarifications**: We expanded and reorganized the Methodology section to provide a clearer and more self-contained description of the normalcy score, and we added additional intuition to improve readability.

- **Experimental protocol and reproducibility**: We added details on the datasets, preprocessing steps, training/testing protocol, and implementation choices to facilitate reproducibility.

- **Additional baselines**: We added intermediate baselines that isolate modeling choices, including a homoscedastic variant (NS\_hom) and a linear variance-estimation baseline (Altman method), and discussed what they reveal about the gains of heteroscedasticity and uncertainty modeling.

- **Additional analyses**: We strengthened result interpretation with additional uncertainty-focused analyses, including a comparison between HDI-length filtering and a context-only Isolation Forest filtering baseline, and we reported interpretable posterior summaries such as exceedance probabilities (e.g., \(P(NS>2)\)) to flag borderline cases beyond point estimates in the medical dataset.

Finally, regarding the concern about limited experimental evaluation (Reviewer KHaV), we would like to stress that, to the best of our knowledge, this is the first work on conditional anomaly detection that reports results on a real-world medical dataset (aorta dataset), in addition to previously studied pseudo-synthetic settings (i.e., anomaly injection on datasets traditionally used for regression).
We thank the reviewers again for their valuable feedback.

---

### Decision · Action_Editor_8AQS · 2026-01-16

**Recommendation:** Accept as is

**Audience:**

Yes

**Audience Explanation:**

Yes -- anomaly detection is an established problem with significant interest and novel methods in this area certainly would be of interest to some members of TMLR's audience.

**Claims And Evidence:**

Yes

**Claims Explanation:**

The reviewers universally agree that the claims made in the manuscript -- in particular, after revision following the discussion period -- are supported by accurate, convincing, and clear evidence, both through well-motivated discussion as well as empirical study.

During the initial review period, some reviewers did express some minor concerns regarding the strength of the claims in the paper, but after some discussion and revision, these concerns were overcome.